# Fast Policy Extragradient Methods for Competitive Games with Entropy Regularization

**Shicong Cen**
Carnegie Mellon University
shicongc@andrew.cmu.edu

**Yuting Wei**
University of Pennsylvania
ytwei@wharton.upenn.edu

**Yuejie Chi**
Carnegie Mellon University
yuejiechi@cmu.edu

## Abstract

This paper investigates the problem of computing the equilibrium of competitive games, which is often modeled as a constrained saddle-point optimization problem with probability simplex constraints. Despite recent efforts in understanding the last-iterate convergence of extragradient methods in the unconstrained setting, the theoretical underpinnings of these methods in the constrained settings, especially those using multiplicative updates, remain highly inadequate, even when the objective function is bilinear. Motivated by the algorithmic role of entropy regularization in single-agent reinforcement learning and game theory, we develop provably efficient extragradient methods to find the quantal response equilibrium (QRE)—which are solutions to zero-sum two-player matrix games with entropy regularization—at a linear rate. The proposed algorithms can be implemented in a decentralized manner, where each player executes symmetric and multiplicative updates iteratively using its own payoff without observing the opponent's actions directly. In addition, by controlling the knob of entropy regularization, the proposed algorithms can locate an approximate Nash equilibrium of the unregularized matrix game at a sublinear rate without assuming the Nash equilibrium to be unique. Our methods also lead to efficient policy extragradient algorithms for solving entropy-regularized zero-sum Markov games at a linear rate. All of our convergence rates are nearly dimension-free, which are independent of the size of the state and action spaces up to logarithm factors, highlighting the positive role of entropy regularization for accelerating convergence.

## 1   Introduction

Finding the equilibrium of competitive games, which can be viewed as constrained saddle-point optimization problems with probability simplex constraints, lies at the heart of modern machine learning and decision making paradigms such as Generative Adversarial Networks (GANs) (Goodfellow et al., 2014), competitive reinforcement learning (RL) (Littman, 1994), game theory (Shapley, 1953), adversarial training (Mertikopoulos et al., 2018b), to name a few.

In this paper, we study one of the most basic forms of competitive games, namely two-player zero-sum games, in both the matrix setting and the Markov setting. Our goal is to find the equilibrium policies of both players in an *independent* and *decentralized* manner (Daskalakis et al., 2020; Wei et al., 2021a) with guaranteed *last-iterate convergence*. Namely, each player will execute symmetric and independent updates iteratively using its own payoff without observing the opponent's actions directly, and the final policies of the iterative process should be a close approximation to the equilibrium up to any prescribed precision. This kind of algorithms is more advantageous and versatile especially in federated environments, as it requires neither prior coordination between the players like two-timescale algorithms, nor a central controller to collect and disseminate the policies of all the players, which are often unavailable due to privacy constraints.

35th Conference on Neural Information Processing Systems (NeurIPS 2021).

## 1.1 Last-iterate convergence in competitive games

In recent years, there have been significant progresses in understanding the last-iterate convergence of simple iterative algorithms for *unconstrained* saddle-point optimization, where one is interested in bounding the sub-optimality of the last iterate of the algorithm, rather than say, the ergodic iterate — which is the average of all the iterations — that are commonly studied in the earlier literature. This shift of focus is motivated, for example, by the infeasibility of averaging large machine learning models in training GANs (Goodfellow et al., 2014). While vanilla Gradient Descent / Ascent (GDA) may diverge or cycle even for bilinear matrix games (Daskalakis et al., 2018), quite remarkably, small modifications lead to guaranteed last-iterate convergence to the equilibrium in a non-asymptotic fashion. A flurry of algorithms is proposed, including Optimistic Gradient Descent Ascent (OGDA) (Rakhlin and Sridharan, 2013; Daskalakis and Panageas, 2018b; Wei et al., 2021b), predictive updates (Yadav et al., 2017), implicit updates (Liang and Stokes, 2019), and more. Several unified analyses of these algorithms have been carried out (see, e.g. Mokhtari et al. (2020a); Liang and Stokes (2019) and references therein), where these methods in principle all make clever extrapolation of the local curvature in a predictive manner to accelerate convergence. With slight abuse of terminology, in this paper, we refer to this ensemble of algorithms as extragradient methods (Korpelevich, 1976; Tseng, 1995; Mertikopoulos et al., 2018a; Harker and Pang, 1990).

However, saddle-point optimization in the *constrained setting*, which includes competitive games as a special case, remains largely under-explored even for bilinear matrix games. While it is possible to reformulate constrained bilinear games to unconstrained ones using softmax parameterization of the probability simplex, this approach falls short of preserving the bilinear structure and convex-concave properties in the original problem, which are crucial to the convergence of gradient methods. Therefore, there is a strong necessity of understanding and developing improved extragradient methods in the constrained setting. Daskalakis and Panageas (2018a) proposed the optimistic variant of the multiplicative weight updates (MWU) method (Arora et al., 2012) – which is extremely natural and popular for optimizing over probability simplexes – called Optimistic Multiplicative Weight Updates (OMWU), and established the asymptotic last-iterate convergence of OMWU for matrix games. Very recently, Wei et al. (2021b) established non-asymptotic last-iterate convergences of OMWU. However, these last-iterate convergence results require the Nash equilibrium to be unique, and cannot be applied to problems with multiple Nash equilibria.

## 1.2 Our contributions

Motivated by the algorithmic role of entropy regularization in single-agent RL (Neu et al., 2017; Geist et al., 2019; Cen et al., 2020) as well as its wide use in game theory to account for imperfect and noisy information (McKelvey and Palfrey, 1995; Savas et al., 2019), we initiate the design and analysis of extragradient algorithms using *multiplicative updates* for finding the quantal response equilibrium (QRE), which are solutions to competitive games with entropy regularization (McKelvey and Palfrey, 1995). While finding QRE is of interest in its own right, by controlling the knob of entropy regularization, the QRE provides a close approximation to the Nash equilibrium (NE), and in turn acts as a smoothing scheme for finding the NE. Our contributions are summarized below.

• *Near dimension-free last-iterate convergence to QRE of entropy-regularized matrix games.* We propose two policy extragradient algorithms to solve entropy-regularized matrix games, namely the Predictive Update (PU) and OMWU methods, where both players execute symmetric and multiplicative updates without knowing the entire payoff matrix nor the opponent's actions. Encouragingly, we show that the last iterate of the proposed algorithms converges to the unique QRE at a linear rate that is almost independent of the size of the action spaces. Roughly speaking, to find an $\epsilon$-optimal QRE in terms of Kullback-Leibler (KL) divergence, it takes no more than $\widetilde{O}\left(\frac{1}{\eta\tau}\log\left(\frac{1}{\epsilon}\right)\right)$ iterations, where $\widetilde{O}(\cdot)$ hides logarithmic dependencies. Here, $\tau$ is the regularization parameter, and $\eta$ is the learning rate of both players. Maximizing the learning rate, the iteration complexity is bounded by $\widetilde{O}\left((1 + \|A\|_\infty/\tau)\log(1/\epsilon)\right)$, where $\|A\|_\infty = \max_{i,j}|A_{i,j}|$ is the $\ell_\infty$ norm of the payoff matrix $A$.

• *Last-iterate convergence to $\epsilon$-NE of unregularized matrix games without uniqueness assumption.* The QRE provides an accurate approximation to the NE by setting the entropy regularization $\tau$ sufficiently small, therefore our result directly translates to finding a NE with last-iterate convergence guarantee. Roughly speaking, to find an $\epsilon$-NE (Zhang et al., 2020, Definition 2.1), it takes no more than $\widetilde{O}\left(1 + \frac{\|A\|_\infty}{\epsilon}\right)$ iterations with optimized learning rates, which is again independent of the size

| Equilibrium type | Method | Convergence rate | Dimension-free | Require unique NE |
|---|---|---|---|---|
| $\epsilon$-QRE | PU & OMWU **(this work)** | linear | yes | n/a |
| $\epsilon$-NE | OMWU (Daskalakis and Panageas, 2018a) | asymptotic | no | yes |
| | OMWU (Wei et al., 2021b) | sublinear + linear | no | yes |
| | PU & OMWU **(this work)** | sublinear | yes | no |

Table 1: Comparisons of last-iterate convergence of the proposed entropy-regularized PU and OMWU methods with prior results for finding $\epsilon$-QRE or $\epsilon$-NE of competitive matrix games. We note that the convergence rates of unregularized OMWU established in Wei et al. (2021b) are problem-dependent, and scale at least polynomially on the size of the action spaces. Desirable features in the last two columns are highlighted in blue.

of the action spaces up to logarithmic factors. Unlike prior literature (Daskalakis and Panageas, 2018a; Wei et al., 2021b), our last-iterate convergence guarantee does not require the NE to be unique.

• *Extensions to two-player zero-sum Markov games.* By connecting value iteration with matrix games, we propose a policy extragradient method for solving infinite-horizon discounted entropy-regularized zero-sum Markov games, which finds an $\epsilon$-optimal minimax soft Q-function—in terms of $\ell_\infty$ error—in at most $\widetilde{O}\left(\frac{1}{\tau(1-\gamma)^2}\log^2\left(\frac{1}{\epsilon}\right)\right)$ iterations, where $\gamma \in (0,1)$ is the discount factor.

To the best of our knowledge, our paper is the first that develops policy extragradient algorithms for solving entropy-regularized competitive games with multiplicative updates and dimension-free linear last-iterate convergence, and demonstrates entropy regularization as a smoothing technique to find $\epsilon$-NE without the uniqueness assumption. Table 1 provides detailed comparisons of the proposed methods with prior arts for solving matrix games. Our results highlight the positive role of entropy regularization for accelerating convergence and safeguarding against imperfect information in competitive games. We defer the complete proof of our results to Cen et al. (2021).

### 1.3 Related works

Our work lies at the intersection of saddle-point optimization, game theory, and reinforcement learning. In what follows, we discuss a few topics that are closely related to ours.

**Unregularized matrix game.** Freund and Schapire (1999) showed that many standard methods such as GDA and MWU have a converging average duality gap at the rate of $O(1/\sqrt{T})$, which is improved to $O(1/T)$ by considering optimistic variants of these methods, such as OGDA and OMWU (Rakhlin and Sridharan, 2013; Daskalakis et al., 2011; Syrgkanis et al., 2015). However, the last-iterate convergence of these methods are less understood until recently (Daskalakis and Panageas, 2018a; Wei et al., 2021b). In particular, under the assumption that the NE is unique for the unregularized matrix game, Daskalakis and Panageas (2018a) showed the asymptotic convergence of the last iterate of OMWU to the unique equilibrium, and Wei et al. (2021b) showed the last iterate of OMWU achieves a linear rate of convergence after an initial phase of sublinear convergence, however the rates therein can be highly pessimistic in terms of the problem dimension, while our rate for entropy-regularized OMWU is dimension-free up to logarithmic factors.

**Saddle-point optimization.** Considerable progress has been made towards understanding OGDA and extragradient (EG) methods in the unconstrained convex-concave saddle-point optimization with general objective functions (Mokhtari et al., 2020a,b; Nemirovski, 2004; Liang and Stokes, 2019). However, the last-iterate convergence of constrained convex-concave saddle-point optimization still lacks theoretical understanding in general and most works fall short of characterizing a finite-time convergence result. In particular, Mertikopoulos et al. (2018a) demonstrated the asymptotic last-iterate convergence of EG, and Hsieh et al. (2019) investigated similar questions for single-call EG algorithms. Lei et al. (2021) showed that OMWU converges to the equilibrium locally without

an explicit rate. Wei et al. (2021b) showed that the last-iterate of OGDA converges linearly for strongly-convex strongly-concave constrained saddle-point optimization with an explicit rate.

**Entropy regularization in RL and games.** In single-agent RL, the role of entropy regularization as an algorithmic mechanism to encourage exploration and accelerate convergence has been investigated extensively (Neu et al., 2017; Geist et al., 2019; Mei et al., 2020; Cen et al., 2020; Lan, 2021; Zhan et al., 2021). Turning to the game setting, entropy regularization is used to account for imperfect information in the seminal work of McKelvey and Palfrey (1995) that introduced the QRE, and a few representative works on entropy and more general regularizations in games include Savas et al. (2019); Hofbauer and Sandholm (2002); Mertikopoulos and Sandholm (2016).

**Zero-sum Markov games.** There have been a significant recent interest in developing provably efficient self-play algorithms for Markov games, including model-based algorithms (Perolat et al., 2015; Zhang et al., 2020), value-based algorithms (Bai and Jin, 2020; Xie et al., 2020), and policy-based algorithms (Daskalakis et al., 2020; Wei et al., 2021a; Zhao et al., 2021). The iteration complexities in prior works (Perolat et al., 2015; Daskalakis et al., 2020; Wei et al., 2021a; Zhao et al., 2021) depend on various notions of concentrability coefficient and therefore can scale quite pessimistically with the problem dimension. Our approach can be regarded as a policy-based algorithm to approximate value iteration, which can be implemented in a decentralized manner with symmetric and multiplicative updates from both players, and the iteration complexity is almost independent of the size of the state-action space.

**Notation.** We denote by $\Delta(\mathcal{A})$ the probability simplex over the set $\mathcal{A}$. We overload the functions such as $\log(\cdot)$ and $\exp(\cdot)$ to take vector inputs with the understanding that the function is applied in an entrywise manner. For instance, given any vector $z = [z_i]_{1 \le i \le n} \in \mathbb{R}^n$, the notation $\exp(z)$ denotes $\exp(z) := [\exp(z_i)]_{1 \le i \le n}$; other functions are defined analogously. Given two probability distributions $\mu$ and $\mu'$ over $\mathcal{A}$, the KL divergence from $\mu'$ to $\mu$ is defined by $\mathsf{KL}\big(\mu \, \| \, \mu'\big) := \sum_{a \in \mathcal{A}} \mu(a) \log \frac{\mu(a)}{\mu'(a)}$. Given a matrix $A$, $\|A\|_\infty$ is used to denote entrywise maximum norm, namely, $\|A\|_\infty = \max_{i,j} |A_{i,j}|$. The all-one vector is denoted as $\mathbf{1}$.

# 2 Zero-sum matrix games with entropy regularization

We first consider a two-player zero-sum game with bilinear objective and probability simplex constraints, and demonstrate the positive role of entropy regularization in solving this problem. Throughout this paper, let $\mathcal{A} = \{1, \dots, m\}$ and $\mathcal{B} = \{1, \dots, n\}$ be the action spaces of each player.

## 2.1 Background and problem formulation

**Zero-sum two-player matrix game.** The focal point of this subsection is a constrained two-player zero-sum matrix game, which can be formulated as the following min-max problem (or saddle point optimization problem):

$$\max_{\mu \in \Delta(\mathcal{A})} \min_{\nu \in \Delta(\mathcal{B})} f(\mu, \nu) := \mu^\top A \nu, \tag{1}$$

where $A \in \mathbb{R}^{m \times n}$ denotes the payoff matrix, $\mu \in \Delta(\mathcal{A})$ and $\nu \in \Delta(\mathcal{B})$ stand for the mixed/randomized policies of each player, defined respectively as distributions over the probability simplex $\Delta(\mathcal{A})$ and $\Delta(\mathcal{B})$. A pair of policies $(\mu^\star, \nu^\star)$ is said to be a *Nash equilibrium (NE)* of (1) if $f(\mu^\star, \nu) \ge f(\mu^\star, \nu^\star) \ge f(\mu, \nu^\star)$     for all $(\mu, \nu) \in \Delta(\mathcal{A}) \times \Delta(\mathcal{B})$.

**Entropy-regularized zero-sum two-player matrix game.** There is no shortage of scenarios where the payoff matrix $A$ might not be known perfectly. In an attempt to accommodate imperfect knowledge of $A$, McKelvey and Palfrey (1995) proposed a seminal extension to the Nash equilibrium called the *quantal response equilibrium (QRE)* when the payoffs are perturbed by Gumbel-distributed noise. Formally, this amounts to solving the following matrix game with entropy regularization (Mertikopoulos and Sandholm, 2016):

$$\max_{\mu \in \Delta(\mathcal{A})} \min_{\nu \in \Delta(\mathcal{B})} f_\tau(\mu, \nu) := \mu^\top A \nu + \tau \mathcal{H}(\mu) - \tau \mathcal{H}(\nu), \tag{2}$$

where $\mathcal{H}(\pi) = -\sum_i \pi_i \log(\pi_i)$ denotes the Shannon entropy of a distribution $\pi$, and $\tau \ge 0$ is the regularization parameter. As is well known, the optimal solution $(\mu_\tau^\star, \nu_\tau^\star)$ to (2), dubbed as the QRE,

is unique whenever $\tau > 0$ (due to the presence of strong concavity/convexity), which satisfies the following fixed point equations:

$$\begin{cases} \mu_\tau^\star(a) = \frac{\exp([A\nu_\tau^\star]_a/\tau)}{\sum_{a=1}^m \exp([A\nu_\tau^\star]_a/\tau)} \propto \exp([A\nu_\tau^\star]_a/\tau), & \text{for all } a \in \mathcal{A}, \\ \nu_\tau^\star(b) = \frac{\exp(-[A^\top \mu_\tau^\star]_b/\tau)}{\sum_{b=1}^n \exp(-[A^\top \mu_\tau^\star]_b/\tau)} \propto \exp(-[A^\top \mu_\tau^\star]_b/\tau), & \text{for all } b \in \mathcal{B}. \end{cases} \tag{3}$$

**Goal.** We aim to efficiently compute the QRE of the entropy-regularized matrix game in a decentralized manner, and investigate how an efficient solver of QRE can be leveraged to find a NE of the unregularized matrix game (1). Namely, we only assume access to "first-order information" as opposed to full knowledge of the payoff matrix $A$ or the actions of the opponent. The information received by each player is formally described in the following sampling oracle.

**Definition 1** (Sampling oracle for matrix games). *For any policy pair $(\mu, \nu)$ and payoff matrix $A$, the sampling oracle returns the exact values of $\mu^\top A$ and $A\nu$.*

**Additional notation.** For notational convenience, we let $\zeta$ represent the concatenation of $\mu \in \mathbb{R}^{|\mathcal{A}|}$ and $\nu \in \mathbb{R}^{|\mathcal{B}|}$, namely, $\zeta = (\mu, \nu)$. The solution to (2), which is specified in (3), is denoted by $\zeta_\tau^\star = (\mu_\tau^\star, \nu_\tau^\star)$. For any $\zeta = (\mu, \nu)$ and $\zeta' = (\mu', \nu')$, we shall often abuse the notation and let $\mathsf{KL}(\zeta \,\|\, \zeta') = \mathsf{KL}(\mu \,\|\, \mu') + \mathsf{KL}(\nu \,\|\, \nu')$. The duality gap of the entropy-regularized matrix game (2) at $\zeta = (\mu, \nu)$ is defined as $\mathsf{DualGap}_\tau(\zeta) = \max_{\mu' \in \Delta(\mathcal{A})} f_\tau(\mu', \nu) - \min_{\nu' \in \Delta(\mathcal{B})} f_\tau(\mu, \nu')$ which is clearly nonnegative and $\mathsf{DualGap}_\tau(\zeta_\tau^\star) = 0$. Similarly, let the optimality gap of the entropy-regularized matrix game (2) at $\zeta = (\mu, \nu)$ be $\mathsf{OptGap}(\zeta) = \left| f_\tau(\mu, \nu) - f_\tau(\mu_\tau^\star, \nu_\tau^\star) \right|$.

## 2.2 Proposed extragradient methods: PU and OMWU

To begin, assume we are given a pair of policies $z_1 \in \Delta(\mathcal{A})$, $z_2 \in \Delta(\mathcal{B})$ employed by each player respectively. If we proceed with fictitious play, i.e. player 1 (resp. player 2) aims to optimize its own policy by assuming the opponent's policy is fixed as $z_2$ (resp. $z_1$), the saddle-point optimization problem (2) is then decoupled into two independent min/max optimization problems:

$$\max_{\mu \in \Delta(\mathcal{A})} \mu^\top A z_2 + \tau \mathcal{H}(\mu) - \tau \mathcal{H}(z_2) \qquad \text{and} \qquad \min_{\nu \in \Delta(\mathcal{B})} z_1^\top A \nu + \tau \mathcal{H}(z_1) - \tau \mathcal{H}(\nu),$$

which are naturally solved via mirror descent/ascent with KL divergence. Specifically, one step of mirror descent/ascent takes the form

$$\begin{cases} \mu^{(t+1)}(a) \propto \mu^{(t)}(a)^{1-\eta\tau} \exp(\eta[Az_2]_a), & \text{for all } a \in \mathcal{A}, \\ \nu^{(t+1)}(b) \propto \nu^{(t)}(b)^{1-\eta\tau} \exp(-\eta[A^\top z_1]_b), & \text{for all } b \in \mathcal{B}, \end{cases} \tag{4}$$

where $\eta$ is the learning rate. The above update rule forms the basis of our algorithm design.

**Motivation: a form of implicit updates with linear convergence.** It turns out, if we could select the policy pair $(z_1, z_2) = \zeta^{(t+1)} := (\mu^{(t+1)}, \nu^{(t+1)})$ as the ones to be taken in the future, and call the resulting update rule as the Implicit Update (IU) method:

$$\text{Implicit Update:} \qquad \begin{cases} \mu^{(t+1)}(a) \propto \mu^{(t)}(a)^{1-\eta\tau} \exp(\eta[A\nu^{(t+1)}]_a), & \text{for all } a \in \mathcal{A}, \\ \nu^{(t+1)}(b) \propto \nu^{(t)}(b)^{1-\eta\tau} \exp(-\eta[A^\top \mu^{(t+1)}]_b), & \text{for all } b \in \mathcal{B}. \end{cases} \tag{5}$$

Though unrealistic — since it uses the future updates — it leads to a one-step convergence to the QRE when $\eta = 1/\tau$ (see the optimality condition in (3)). Encouragingly, we have the following linear convergence guarantee of IU when adopting a general learning rate.

**Proposition 1** (Linear convergence of IU). *Assume $0 < \eta \leq 1/\tau$, then for all $t \geq 0$, the iterates $\zeta^{(t)} := (\mu^{(t)}, \nu^{(t)})$ of the IU method in (5) satisfy $\mathsf{KL}(\zeta_\tau^\star \,\|\, \zeta^{(t)}) \leq (1 - \eta\tau)^t \mathsf{KL}(\zeta_\tau^\star \,\|\, \zeta^{(0)})$.*

In words, the IU method achieves an appealing linear rate of convergence that is independent of the problem dimension. Motivated by this observation, we seek to design algorithms where the policies $(z_1, z_2)$ employed in (4) serve as good predictions of $(\mu^{(t+1)}, \nu^{(t+1)})$, such that the resulting algorithms are both practical and retain the appealing convergence rate of IU.

**Proposed algorithms.** We propose two extragradient algorithms for solving the entropy-regularized matrix game, namely the *Predictive Update (PU)* method and the *Optimistic Multiplicative Weights*

| **Algorithm 1:** The PU method | **Algorithm 2:** The OMWU method |
|---|---|
| **1 initialization:** $\mu^{(0)}, \nu^{(0)}$. | **1 initialization:** $\mu^{(0)} = \bar{\mu}^{(0)}, \nu^{(0)} = \bar{\nu}^{(0)}$. |
| **2 for** $t = 0, 1, 2, \cdots$ **do** | **2 for** $t = 0, 1, 2, \cdots$ **do** |
| **3**    Update $\bar{\mu}$ and $\bar{\nu}$ according to | **3**    Update $\bar{\mu}$ and $\bar{\nu}$ according to |
| $$\begin{cases} \bar{\mu}^{(t+1)}(a) \propto \mu^{(t)}(a)^{1-\eta\tau} \exp(\eta[A\nu^{(t)}]_a), \\ \bar{\nu}^{(t+1)}(b) \propto \nu^{(t)}(b)^{1-\eta\tau} \exp(-\eta[A^\top\mu^{(t)}]_b). \end{cases}$$ | $$\begin{cases} \bar{\mu}^{(t+1)}(a) \propto \mu^{(t)}(a)^{1-\eta\tau} \exp(\eta[A\bar{\nu}^{(t)}]_a), \\ \bar{\nu}^{(t+1)}(b) \propto \nu^{(t)}(b)^{1-\eta\tau} \exp(-\eta[A^\top\bar{\mu}^{(t)}]_b). \end{cases}$$ |
| **4**    Update $\mu$ and $\nu$ according to | **4**    Update $\mu$ and $\nu$ according to |
| $$\begin{cases} \mu^{(t+1)}(a) \propto \mu^{(t)}(a)^{1-\eta\tau} \exp(\eta[A\bar{\nu}^{(t+1)}]_a), \\ \nu^{(t+1)}(b) \propto \nu^{(t)}(b)^{1-\eta\tau} \exp(-\eta[A^\top\bar{\mu}^{(t+1)}]_b). \end{cases}$$ | $$\begin{cases} \mu^{(t+1)}(a) \propto \mu^{(t)}(a)^{1-\eta\tau} \exp(\eta[A\bar{\nu}^{(t+1)}]_a), \\ \nu^{(t+1)}(b) \propto \nu^{(t)}(b)^{1-\eta\tau} \exp(-\eta[A^\top\bar{\mu}^{(t+1)}]_b). \end{cases}$$ |

*Update (OMWU)* method, the latter adapted from Rakhlin and Sridharan (2013); Daskalakis et al. (2011). Detailed procedures can be found in Algorithm 1 and Algorithm 2, respectively. On a high level, both algorithms maintain two intertwined sequences $\{(\mu^{(t)}, \nu^{(t)})\}_{t \geq 0}$ and $\{(\bar{\mu}^{(t)}, \bar{\nu}^{(t)})\}_{t \geq 0}$, and in each iteration $t = 0, 1, \ldots$, proceed in two steps:

• The midpoint $(\bar{\mu}^{(t+1)}, \bar{\nu}^{(t+1)})$ serves as a prediction of $(\mu^{(t+1)}, \nu^{(t+1)})$ by running one step of mirror descent / ascent (cf. (4)) from either $(z_1, z_2) = (\mu^{(t)}, \nu^{(t)})$ (for PU) or $(z_1, z_2) = (\bar{\mu}^{(t)}, \bar{\nu}^{(t)})$ (for OMWU).

• The update of $(\mu^{(t+1)}, \nu^{(t+1)})$ then mimics the implicit update (5) using the prediction $(\bar{\mu}^{(t+1)}, \bar{\nu}^{(t+1)})$ obtained above.

When the proposed algorithms converge, both $(\mu^{(t)}, \nu^{(t)})$ and $(\bar{\mu}^{(t)}, \bar{\nu}^{(t)})$ converge to the same point. The two players are completely symmetric and adopt the same learning rate, and require *only* first-order information provided by the sampling oracle. While the two algorithms resemble each other in many aspects, a key difference lies in the query and use of the sampling oracle: in each iteration, OMWU makes a single call to the sampling oracle for gradient evaluation, while PU calls the sampling oracle twice. It is worth noting that, when $\tau = 0$ (i.e., no entropy regularization is enforced), the OMWU method in Algorithm 2 reduces to the method analyzed in Rakhlin and Sridharan (2013); Daskalakis and Panageas (2018a); Wei et al. (2021b) without entropy regularization.

**Remark 1.** *It is worth highlighting that the proposed algorithms are* different *from Mertikopoulos et al. (2018a), as the extragradient is only applied to the bilinear term but not the entropy regularization term. This seemingly small, but important, difference leads to a more concise closed-form update rule and a cleaner analysis, as shall be seen momentarily.*

### 2.3 Performance guarantees

We are now positioned to present our main theorem concerning the last-iterate convergence of PU and OMWU for solving (2).

**Theorem 1** (Last-iterate convergence of PU and OMWU). *Suppose that the learning rates $\eta = \eta_{\mathsf{PU}}$ of PU in Algorithm 1 and $\eta = \eta_{\mathsf{OMWU}}$ of OMWU in Algorithm 2 satisfy*

$$0 < \eta_{\mathsf{PU}} \leq \frac{1}{\tau + 2\|A\|_\infty}, \quad and \quad 0 < \eta_{\mathsf{OMWU}} \leq \min\left\{\frac{1}{2\tau + 2\|A\|_\infty}, \frac{1}{4\|A\|_\infty}\right\}. \quad (6)$$

*Then for any $t \geq 0$, the iterates $\zeta^{(t)} = (\mu^{(t)}, \nu^{(t)})$ and $\bar{\zeta}^{(t)} = (\bar{\mu}^{(t)}, \bar{\nu}^{(t)})$ of PU and OMWU achieve*

• **Linear convergence of policies in KL divergence and entrywise log-ratios:**

$$\max\left\{\mathsf{KL}\big(\zeta_\tau^\star \,\|\, \zeta^{(t)}\big), \tfrac{1}{2}\mathsf{KL}\big(\zeta_\tau^\star \,\|\, \bar{\zeta}^{(t+1)}\big)\right\} \leq (1 - \eta\tau)^t \mathsf{KL}\big(\zeta_\tau^\star \,\|\, \zeta^{(0)}\big), \quad (7a)$$

$$\left\|\log\frac{\zeta^{(t)}}{\zeta_\tau^\star}\right\|_\infty \leq 2(1 - \eta\tau)^t \left\|\log\frac{\zeta^{(0)}}{\zeta_\tau^\star}\right\|_\infty + \frac{8\|A\|_\infty}{\tau}(1 - \eta\tau)^{t/2}\mathsf{KL}\big(\zeta_\tau^\star \,\|\, \zeta^{(0)}\big)^{1/2}. \quad (7b)$$

• **Linear convergence of values in optimality and duality gaps:**

$$\mathsf{OptGap}_\tau(\bar{\zeta}^{(t)}) \leq \eta^{-1} \cdot \frac{1}{1 - (\tau + \|A\|_\infty)\eta} \cdot \frac{(1 - \eta\tau)^t}{1 - (1 - \eta\tau)^t} \mathsf{KL}\big(\zeta_\tau^\star \,\|\, \zeta^{(0)}\big), \qquad (7c)$$

$$\mathsf{DualGap}_\tau(\bar{\zeta}^{(t)}) \leq \big(\eta^{-1} + 2\tau^{-1}\|A\|_\infty^2\big)(1 - \eta\tau)^{t-1}\mathsf{KL}\big(\zeta_\tau^\star \,\|\, \zeta^{(0)}\big). \qquad (7d)$$

**Remark 2.** *Setting $\mu^{(0)}$ and $\nu^{(0)}$ to be uniform policies leads to a universal bound*

$$\mathsf{KL}\big(\zeta_\tau^\star \,\|\, \zeta^{(0)}\big) = \log|\mathcal{A}| + \log|\mathcal{B}| - \mathcal{H}(\mu_\tau^\star) - \mathcal{H}(\nu_\tau^\star) \leq \log|\mathcal{A}| + \log|\mathcal{B}|.$$

**Remark 3.** *Similar results continue to hold even when the two players use different regularization parameters $\tau_\mu, \tau_\nu > 0$ in (2), as long as the regularization parameter $\tau$ is replaced by $\max\{\tau_\mu, \tau_\nu\}$ in the upper bounds of the learning rate, and the contraction parameter is replaced by $1 - \min\{\tau_\mu, \tau_\nu\}\eta$.*

Theorem 1 characterizes the convergence of the *last-iterates* $\zeta^{(t)}$ and $\bar{\zeta}^{(t)}$ of PU and OMWU as long as the learning rate lies within the specified ranges. While PU doubles the number of calls to the sampling oracle, it also allows roughly as large as twice the learning rate compared with OMWU (cf. (6)). Compared with the vast literature analyzing the average-iterate performance of variants of extragradient methods, our results contribute towards characterizing the last-iterate convergence of multiplicative update methods in the presence of entropy regularization and simplex constraints, which to the best of our knowledge, are the first of its kind. Several remarks are in order.

**Linear convergence to QRE.** To achieve an $\epsilon$-accurate estimate of the QRE in terms of the KL divergence, the bound (7a) tells that it is sufficient to take

$$\frac{1}{\eta\tau} \log\left(\frac{\log|\mathcal{A}| + \log|\mathcal{B}|}{\epsilon}\right)$$

iterations using either PU or OMWU. Notably, this iteration complexity does not depend on any hidden constants and only depends double logarithmically on the cardinality of action spaces, which is almost dimension-free. Maximizing the learning rate, the iteration complexity is bounded by $(1 + \|A\|_\infty/\tau)\log(1/\epsilon)$ (modulo log factors), which only depends on the ratio $\|A\|_\infty/\tau$.

**Entrywise error of the policy log-ratios.** Both PU and OMWU enjoy strong entrywise guarantees in the sense we can guarantee the convergence of the $\ell_\infty$ norm of the log-ratios between the learned policy pair and the QRE at the same dimension-free linear rate (cf. (7b)), which suggests the policy pair converges in a somewhat uniform manner across the entire action space.

**Linear convergence of optimality and duality gaps.** Our theorem also establishes the last-iterate convergence of the game values in terms of the optimality gap (cf. (7c)) and the duality gap (cf. (7d)) for both PU and OMWU. In particular, as will be seen, bounding the optimality gap of matrix games turns out to be the key enabler for generalizing our algorithms to Markov games, and bounding the duality gap allows to directly translate our results to finding a NE of unregularized matrix games.

**Last-iterate convergence to approximate NE.** The entropy-regularized matrix game can be thought as a smooth surrogate of the unregularized matrix game (1); in particular, it is possible to find an $\epsilon$-NE by setting $\tau$ sufficiently small in (2). According to (Zhang et al., 2020, Definition 2.1), a policy pair $\zeta = (\mu, \nu)$ is an $\epsilon$-NE if it satisfies $\mathsf{DualGap}(\zeta) := \max_{\mu' \in \Delta(\mathcal{A})} f(\mu', \nu) - \min_{\nu' \in \Delta(\mathcal{B})} f(\mu, \nu') \leq \epsilon$.

Observe that setting $\tau = \frac{\epsilon/4}{\log|\mathcal{A}| + \log|\mathcal{B}|}$ guarantees that $|f_\tau(\mu, \nu) - f(\mu, \nu)| < \epsilon/4$ uniformly over $(\mu, \nu) \in \Delta(\mathcal{A}) \times \Delta(\mathcal{B})$ in view of the boundedness of the Shannon entropy $\mathcal{H}(\cdot)$. Theorem 7 (cf. (7d)) also ensures that our proposed algorithms find an approximate QRE $\bar{\zeta}^{(T)}$ such that $\mathsf{DualGap}_\tau(\bar{\zeta}^{(T)}) \leq \epsilon/2$ after taking $T = \widetilde{O}\left(\frac{1}{\eta\epsilon}\right)$ iterations, which is no more than $\widetilde{O}\left(1 + \frac{\|A\|_\infty}{\epsilon}\right)$ iterations with optimized learning rates. It follows immediately that

$$\mathsf{DualGap}(\bar{\zeta}^{(T)}) \leq \mathsf{DualGap}_\tau(\bar{\zeta}^{(T)}) + \max_{\mu',\nu'}\left|f_\tau(\mu', \bar{\nu}^{(T)}) - f_\tau(\bar{\mu}^{(T)}, \nu') - (f(\mu', \bar{\nu}^{(T)}) - f(\bar{\mu}^{(T)}, \nu'))\right| \leq \epsilon,$$

and therefore $\bar{\zeta}^{(T)}$ is an $\epsilon$-NE. Intriguingly, unlike prior work (Daskalakis and Panageas, 2018a; Wei et al., 2021b) that analyzed the last-iterate convergence of OMWU in the unregularized setting ($\tau = 0$), our last-iterate convergence does not require the NE of (1) to be unique.

**Rationality.** Another attractive feature of the algorithms developed above is being *rational* (as introduced in Bowling and Veloso (2001)) in the sense that the algorithm returns the best-response

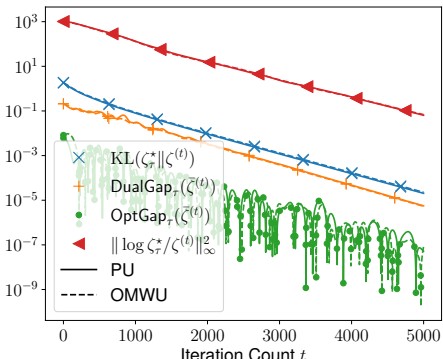 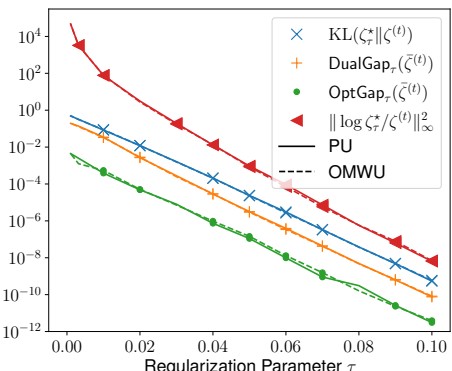

Figure 1: Performance illustration of the PU and OMWU methods for solving entropy-regularized matrix games with $|\mathcal{A}| = |\mathcal{B}| = 100$, where the entries of the payoff matrix $A$ is generated independently from the uniform distribution on $[-1, 1]$. The learning rates are fixed as $\eta = 0.1$. The left panel plots various error metrics of convergence w.r.t. the iteration count with $\tau = 0.01$, while the right panel plots these error metrics at 1000-th iteration with different choices of $\tau$.

policy of one player when the opponent takes any *fixed* stationary policy. More specially, in terms of matrix games, when player 2 sticks to a stationary policy $\nu$, the update of player 1 reduces to $\mu^{(t+1)}(a) \propto \mu^{(t)}(a)^{1-\eta\tau} \exp(\eta[A\nu]_a)$. In this case, Theorem 1 can be established in exactly the same fashion by restricting attention only to the updates of $\mu^{(t)}$.

**No-regret learning of OMWU.** Besides convergence to equilibria, in game-theoretical settings, it is often desirable to design and implement no-regret algorithms, which are capable of providing black-box guarantees over arbitrary sequences played by the opponent (Cesa-Bianchi and Lugosi, 2006; Rakhlin and Sridharan, 2013). Fortunately, it turns out that entropy regularization not only accelerates the convergence, but also enables no-regret learning somewhat "for free": it encourages exploration by putting a positive mass on every action, therefore guards against adversaries. By using a properly chosen learning rate schedule, the proposed OMWU (Algorithm 2) can be further established as a no-regret algorithm; the details can be found in (Cen et al., 2021).

## 3 Zero-sum Markov games with entropy regularization

Leveraging the success of PU and OMWU in solving the entropy-regularized matrix games, this section extends our current analysis to solve the zero-sum two-player Markov game with entropy regularization, which is again formulated as finding the equilibrium of a saddle-point optimization problem. We start by introducing its basic setup, which will be followed by the proposed policy extragradient method with its theoretical guarantees.

### 3.1 Background and problem formulation

We consider a discounted Markov Game (MG) which is defined as $\mathcal{M} = \{\mathcal{S}, \mathcal{A}, \mathcal{B}, P, r, \gamma\}$, with discrete state space $\mathcal{S}$, action spaces of two players $\mathcal{A}$ and $\mathcal{B}$, transition probability $P$, reward function $r : \mathcal{S} \times \mathcal{A} \times \mathcal{B} \to [0, 1]$ and discount factor $\gamma \in [0, 1)$. A policy $\mu : \mathcal{S} \to \Delta(\mathcal{A})$ (resp. $\nu : \mathcal{S} \to \Delta(\mathcal{B})$) defines how player 1 (resp. player 2) reacts to a given state $s$, where the probability of taking action $a \in \mathcal{A}$ (resp. $b \in \mathcal{B}$) is $\mu(a|s)$ (resp. $\nu(b|s)$). The transition probability kernel $P : \mathcal{S} \times \mathcal{A} \times \mathcal{B} \to \Delta(\mathcal{S})$ defines the dynamics of the Markov game, where $P(s'|s, a, b)$ specifies the probability of transiting to state $s'$ from state $s$ when the players take actions $a$ and $b$ respectively.

Motivated by entropy regularization in Markov decision processes (MDP) (Geist et al., 2019), we consider an entropy-regularized variant of MG, where the value function is defined as

$$V_\tau^{\mu,\nu}(s) := \mathbb{E}\left[\sum_{t=0}^{\infty} \gamma^t \left(r(s_t, a_t, b_t) - \tau \log \mu(a_t|s_t) + \tau \log \nu(b_t|s_t)\right) \,\Big|\, s_0 = s\right], \tag{8}$$

where the quantity $\tau \geq 0$ denotes the regularization parameter, and the expectation is evaluated over the randomness of the transition kernel as well as the policies. The regularized Q-function $Q_\tau^{\mu,\nu}$ of a

policy pair $(\mu, \nu)$ is related to $V_\tau^{\mu,\nu}$ as

$$Q_\tau^{\mu,\nu}(s,a,b) = r(s,a,b) + \gamma \mathbb{E}_{s' \sim P(\cdot|s,a,b)} \left[ V_\tau^{\mu,\nu}(s') \right]. \tag{9}$$

We will call $V_\tau^{\mu,\nu}$ and $Q_\tau^{\mu,\nu}$ the *soft value function* and *soft Q-function*, respectively. A policy pair $(\mu_\tau^\star, \nu_\tau^\star)$ is said to be the quantal response equilibrium (QRE) of the entropy-regularized MG, if its value attains the minimax value of the entropy-regularized MG over all states $s \in \mathcal{S}$, i.e.

$$V_\tau^\star(s) = \max_\mu \min_\nu V_\tau^{\mu,\nu}(s) = \min_\nu \max_\mu V_\tau^{\mu,\nu}(s) := V_\tau^{\mu_\tau^\star,\nu_\tau^\star}(s),$$

where $V_\tau^\star$ is called the optimal minimax soft value function, and similarly $Q_\tau^\star := Q_\tau^{\mu_\tau^\star,\nu_\tau^\star}$ is called the optimal minimax soft Q-function.

**Goal.** Our goal is to find the QRE of the entropy-regularized MG in a decentralized manner where the players only observe its own reward without accessing the opponent's actions. By setting the regularization parameter sufficiently small $\tau$, this also allows us to find an approximate NE of the unregularized MG.

### 3.2 From value iteration to policy extragradient methods

**Entropy-regularized value iteration.** It is known that classical dynamic programming approaches such as value iteration can be extended to solve MG (Perolat et al., 2015), where each iteration amounts to solving a series of matrix games for each state. Similar to the single-agent case (Cen et al., 2020), we can extend these approaches to solve the entropy-regularized MG. Setting the stage, let us introduce the per-state Q-value matrix $Q(s) := Q(s, \cdot, \cdot) \in \mathbb{R}^{|\mathcal{A}| \times |\mathcal{B}|}$ for every $s \in \mathcal{S}$, where the element indexed by the action pair $(a, b)$ is $Q(s, a, b)$. Similarly, we define the per-state policies $\mu(s) := \mu(\cdot|s) \in \Delta(\mathcal{A})$ and $\nu(s) := \nu(\cdot|s) \in \Delta(\mathcal{B})$ for both players.

In parallel to the original Bellman operator, we denote the *soft Bellman operator* $\mathcal{T}_\tau$ as

$$\mathcal{T}_\tau(Q)(s,a,b) := r(s,a,b) + \gamma \mathbb{E}_{s' \sim P(\cdot|s,a,b)} \left[ \max_{\mu(s') \in \Delta(\mathcal{A})} \min_{\nu(s') \in \Delta(\mathcal{B})} f_\tau \big( Q(s'); \mu(s'), \nu(s') \big) \right],$$

where for each per-state Q-value matrix $Q(s)$, we introduce an entropy-regularized matrix game in the form of

$$\max_{\mu \in \Delta(\mathcal{A})} \min_{\nu \in \Delta(\mathcal{B})} f_\tau \big( Q(s); \mu(s), \nu(s) \big) := \mu(s)^\top Q(s) \nu(s) - \tau \mathcal{H}(\mu(s)) + \tau \mathcal{H}(\nu(s)).$$

The entropy-regularized value iteration then proceeds as

$$Q^{(t+1)} = \mathcal{T}_\tau(Q^{(t)}), \tag{10}$$

where $Q^{(0)}$ is an initialization. By definition, the optimal minimax soft Q-function obeys $\mathcal{T}_\tau(Q_\tau^\star) = Q_\tau^\star$ and therefore corresponds to the fix point of the soft Bellman operator. Given the above entropy-regularized value iteration, the following lemma states its iterates contract linearly to the optimal minimax soft Q-function at a rate of the discount factor $\gamma$.

**Proposition 2.** *The entropy-regularized value iteration* (10) *converges at a linear rate, i.e.* $\|Q^{(t)} - Q_\tau^\star\|_\infty \leq \gamma^t \|Q^{(0)} - Q_\tau^\star\|_\infty$.

**Approximate value iteration via policy extragradient methods.** Proposition 2 suggests that the optimal minimax soft Q-function of the entropy-regularized MG can be found by solving a series of entropy-regularized matrix games induced by $\{Q^{(t)}\}_{t \geq 0}$ in (10), a task that can be accomplished by adopting the fast extragradient methods developed earlier. To proceed, we first define the following sampling oracle, which makes it rigorous that the proposed algorithm does not require access to the Q-function of the entire MG, but only its own single-agent Q-function when playing against the opponent's policy.

**Definition 2** (Sampling oracle for Markov games). *Given any policy pair* $\mu(s), \nu(s)$ *and Q-value matrix* $Q(s)$ *for any* $s \in \mathcal{S}$, *the sampling oracle returns*

$$[Q(s)\nu(s)]_a = \mathbb{E}_{b \sim \nu(s)} [Q(s,a,b)], \qquad \text{and} \qquad [Q(s)^\top \mu(s)]_b = \mathbb{E}_{a \sim \mu(s)} [Q(s,a,b)]$$

*for any* $a \in \mathcal{A}$ *and* $b \in \mathcal{B}$.

**Algorithm 3:** Policy Extragradient Method for Entropy-regularized Markov Game

---

1 **initialization:** $Q^{(0)} = 0$.
2 **for** $t = 0, 1, 2, \cdots, T_{\text{main}}$ **do**
3     Let $Q^{(t)}$ denote

$$Q^{(t)}(s, a, b) = r(s, a, b) + \gamma \mathbb{E}_{s' \sim P(\cdot|s,a,b)} V^{(t)}(s'). \qquad (11)$$

4     Invoke PU (Algorithm 1) or OMWU (Algorithm 2) for $T_{\text{sub}}$ iterations to solve the following entropy-regularized matrix game for every state $s$, where the initialization is set as uniform distributions:

$$\max_{\mu(s) \in \Delta(\mathcal{A})} \min_{\nu(s) \in \Delta(\mathcal{B})} f_\tau\big(Q^{(t)}(s); \mu(s), \nu(s)\big).$$

    Return the last iterate $\bar{\mu}^{(t,T_{\text{sub}})}(s), \bar{\nu}^{(t,T_{\text{sub}})}(s)$.
5     Set $V^{(t+1)}(s) = f_\tau\big(Q^{(t)}(s); \bar{\mu}^{(t,T_{\text{sub}})}(s), \bar{\nu}^{(t,T_{\text{sub}})}(s)\big)$.

---

Encouragingly, by judiciously setting the number of iterations in both the outer loop (for updating the Q-value matrices) and the inner loop (for updating the QRE of the corresponding Q-value matrix), we are guaranteed to find the QRE of the entropy-regularized MG in a small number of iterations without solving the iteration-varying matrix games exactly, as dictated by the following theorem.

**Theorem 2.** *Assume $|\mathcal{A}| \geq |\mathcal{B}|$ and $\tau \leq 1$. Setting $\eta = \frac{1-\gamma}{2(1+\tau(\log|\mathcal{A}|+1-\gamma))}$, the total iterations (namely, the product $T_{\text{main}} \cdot T_{\text{sub}}$) required for Algorithm 3 to achieve $\big\|Q^{(T_{\text{main}})} - Q_\tau^\star\big\|_\infty \leq \epsilon$ is at most $O\left(\frac{(\log|\mathcal{A}|+1/\tau)}{(1-\gamma)^2}\left(\log\frac{\log|\mathcal{A}|}{(1-\gamma)\epsilon}\right)^2\right)$.*

Theorem 2 ensures that within $\widetilde{O}\left(\frac{1}{\tau(1-\gamma)^2}\log^2\left(\frac{1}{\epsilon}\right)\right)$ iterations, Algorithm 3 finds a pair of policies whose value is close to the optimal minimax soft Q-function $Q_\tau^\star$ in an entrywise manner to a prescribed accuracy $\epsilon$. Remarkably, the iteration complexity is independent of the dimensions of the state space and the action space (up to log factors).

**Remark 4** (Duality gap and solving the unregularized MG). *Solving the entropy-regularized MG provides a viable strategy to find an $\epsilon$-approximate NE of the unregularized MG, where the optimality of a policy pair is typically gauged by the duality gap. Fortunately, this can be achieved similar to the case of matrix games, and we refer interested readers to Cen et al. (2021) for details.*

## 4   Conclusions

This paper develops provably efficient policy extragradient methods (PU and OMWU) for entropy-regularized matrix games and Markov games, whose last iterates are guaranteed to converge linearly to the quantal response equilibrium at a linear rate. Encouragingly, the rate of convergence is independent of the dimension of the problem, i.e. the sizes of the space space and the action space. In addition, the last iterates of the proposed algorithms can also be used to locate Nash equilibria for the unregularized competitive games without assuming the uniqueness of the Nash equilibria by judiciously tuning the amount of regularization. This work opens up interesting opportunities for further investigations of policy extragradient methods for solving competitive games. For example, can we develop a two-time-scale policy extragradient algorithms for Markov games where the Q-function is updated simultaneously with the policy but potentially at a different time scale, using samples, such as in an actor-critic algorithm (Konda and Tsitsiklis, 2000)?

## Acknowledgments and Disclosure of Funding

S. Cen and Y. Chi are supported in part by the grants ONR N00014-18-1-2142 and N00014-19-1-2404, ARO W911NF-18-1-0303, NSF CCF-1901199, CCF-2007911 and CCF-2106778. Y. Wei is supported in part by the NSF grants CCF-2007911, DMS-2147546/2015447 and CCF-2106778.

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
