# OpenReview forum: "Fast Policy Extragradient Methods for Competitive Games with Entropy Regularization"
_NeurIPS.cc/2021/Conference — NeurIPS 2021 Poster_

### Official Review · Reviewer_bKGv · 2021-07-14

**Rating:** 6
**Confidence:** 1

**Summary:**

The paper proposes provably efficient policy extragradient methods for MG, where the last iterates are proven to converge linearly to the QRE of the game. Such a rate of convergence is dimension independent, and the methods can be implemented without further assuming the uniqueness of NE.

**Limitations And Societal Impact:**

Yes.

**Main Review:**

### originality
The paper is original. The algorithms themselves are not original but the finite-time last iterate analysis is novel and technically sound.

### quality
The paper is of good quality.

### clarity
The paper is well written and easy to follow.

### significance
MG is an important type of games and the paper targets on solving MG using QRE as an approximation of NE, without further assuming that NE is unique.

## minor issues
Why entropy regularization is important in Markov games? Can we loose the condition here?

**Time Spent Reviewing:**

1

---

> ### Author Response · Authors · 2021-08-10
> **Thank you very much for the positive evaluation of our paper and the valuable feedback.**
>
> Proposed by [McKelvey and Palfrey, 1995], the entropy-regularized Nash equilibrium is often referred to as the quantal response equilibrium (QRE) when the payoffs are perturbed by Gumbel-distributed noise. Compared to the solution of NE, entropy regularization is frequently adopted to encourage exploration and prevent overfitting, therefore it can be viewed as an important generalization of unregularized Markov games. While solving QRE is of interest in its own right, by controlling the level of entropy regularization, the QRE also provides a close approximation to the NE and often help accelerate the convergence rates, as demonstrated in our work.

---

### Official Review · Reviewer_wNLY · 2021-07-15

**Rating:** 5
**Confidence:** 4

**Summary:**

This paper proposes to leverage entropic regularization and optimistic-type methods to solve zero-sum (ZS) matrix games and ZS Markov games.
The authors begin with ZS matrix games on the simplex: they add entropic regularization, thus transforming this game into a strongly-concave/strongly-concave problem.
Slight variants of Mirror-prox/extragradient and optimistic mirror descent with multiplicative updates are introduced, and the authors show that, for several performance measures, the last-iterate of these methods converge linearly to a solution of the regularized problem thanks to the added strong convexity/concavity.
Their rates are (almost) dimension-free, and do not require the uniqueness of the solution.
As a consequence, they explain how to obtain $\epsilon$-Nash-equilibrium with $O(1/\epsilon)$ iterations by tuning the regularization parameter.
Finally, the authors propose to put to use their results to tackle ZS infinite-horizon Markov games.

**Main Review:**

## Originality and significance

### ZS Matrix Games
- Optimistic methods and their last-iterate convergence have been heavily studied recently.
- Entropic regularization for games seem also to have attracted interest, especially as an alternative solution concept (see references from the submission)
The authors seem to be the first to combine these two principles and to demonstrate the computational benefits of such a match.

### ZS Markov Games
- Optimistic methods have also gained traction for this problem [Wei et al. 2021a, Zhao et al. 2021]
- Entropic regularization has also been proposed in the single-agent MDP case, for instance [Cen et al. 2020]
Again, the authors combine these two notions, and seem almost to be the first to do so.
Indeed, though this may be considered concurrent work (appeared on arXiv 3 months before the NeurIPS deadline), [Zhao et al. 2021] also consider entropic regularization in section 4.1. Though I would expect their rate to be worse than the one of the submission, I am not sure it is the case (their step-size seems bigger...).

## Quality and clarity
The paper seems technically sound and is quite clear.

## Strengths and weaknesses

### Strengths
- The authors propose a simple yet theoretically efficient method to solve ZS matrix games, with little assumption and dimension-free rates.
- They use their method as a subroutine to solve ZS Markov Games efficiently.
+ Good comparison with existing work on OMWU

### Weaknesses
- Lack of (at least theoretical) comparison with other methods to solve ZS Markov games (cf question below)

### Questions
- Could the authors expand on the comparison with the results with  entropic regularization of [Zhao et al. 2021], and more generally compare their work more thoroughly to the litterature on ZS Markov games, in particular with [Wei. et al. 2021a, Zhao et al.2021, Daskalakis et al. 2020] which are already cited in the submission?
- Is is possible, using this approach for ZS Markov games, to have performance guarantees w.r.t. the solution of the unregularized problem? (in the same fashion as lines 255-265 for ZS matrix games, and since the authors seem to imply that this is also the case for ZS Markov games lines 299-300)
- Why is the last-iterate convergence so important for ZS games? In particular, what would be advantages of finding an approximate Nash equilibrium with the method from this paper vs classical averaged Mirror-Prox? (the number of iteration is of the same order with the same dimension-free properties, as far as I understand)

### Typos
- Line 64 of the supplementary material: inequalities -> equalities
- First equality below line 169 in the supplemnetray material: $\mu$ -> $\mu(s)$, $\nu$ -> $\nu(s)$

### Edit after authors' response
Rating changed, see my response below

**Time Spent Reviewing:**

20

---

> ### Author Response · Authors · 2021-08-10
> **Thanks so much for the valuable feedback. Point-to-point responses are provided below.**
>
> ### --- Performance guarantee for unregularized ZS Markov games and comparison with other [Zhao et al. 2021], [Wei et al. 2021a] [Daskalaskis et al. 2020].
>
> While Theorem 2 is stated in terms of regularized $Q$-values, it is also possible to derive duality convergence results for unregularized ZS Markov games and achieve dimension-free rates (up to log factors), as similarly done for the matrix game setup. We will include more details about this extension in our revision. Specifically, our approach achieves $ \mathsf{DualGap} \le \epsilon$ within $\widetilde{O} (\frac{1}{ (1-\gamma)^3 \epsilon})$ iterations.  When comparing with [Zhao et al. 2021], [Wei et al. 2021a] [Daskalaskis et al. 2020] which are proposed to solve the unregularized problem, we note that our algorithm is the only that simultaneously possesses symmetric updates, problem-independent rates, and last-iterate convergence:
>
> 1) [Zhao et al. 2021] follows the approximate policy iteration approach in [Perolat et al. 2015], which switches between solving matrix games and solving single-agent MDPs. Only the latter part (solving single-agent MDPs) involves entropy regularization (following [Cen et al. 2020]) as a means to speed up NPG and the matrix game part is solved via perturbed OMWU established [Rakhlin and Sridharan, 2013]. This is in stark contrast to our approach which directly studies entropy-regularized Markov games where each step solves an entropy-regularized matrix game. In addition, the complexity result in [Zhao et al. 2021] depends on several concentrability coefficients that may be exclusively large. The two players in [Zhao et al. 2021] are also not symmetric, while in our approach the two players perform symmetric updates.
>
> 2) Similarly, in [Daskalaskis et al. 2020], the two players are also not symmetric by using unequal learning rates, and their convergence guarantee is with respect to the averaged iterates rather than the final iterates, which again depends on problem dimensions (via the minimax mismatch coefficient). In fact, it was proposed as an open question on how to leverage extragradient methods to achieve last-iterate convergence, which our works shed some light on.
>
> 3) [Wei et al. 2021a]  proposed a projected ODGA algorithm that admits last-iterate convergence and symmetric updates like ours, however, their convergence rate is quite pessimistic in terms of problem dimensions, where some problem-dependent constant can be excessive (similar to our discussions in Appendix A, since the algorithm in [Wei et al. 2021a] is built on the matrix game algorithm in [Wei et al. 2021b], therefore, suffers from similar dimension-dependence issues).
>
>
> We will make these comparisons more explicit in our revision.
>
>
> ### --- Why is the last-iterate convergence so important for ZS games?
>
> Last-iterate convergence has been deemed as an important feature in game theory, as it better describes the true dynamic of the game evolution. Furthermore, the last-iterate convergence is a stronger notion than average-iterate (or ergodic) convergence, since the former implies the latter but not vice versa (see [Mertikopoulos et al, 2018b] for example). Hence, proving last-iterate convergence is of great interest, and has received considerable attention in the literature, as discussed in Section 1.1 in our paper.
>
>
> ### --- Typos
> Thanks for catch these. We will fix the typos in the revision.
>
> [1] Mertikopoulos, P., Papadimitriou, C., and Piliouras, G. (2018b). Cycles in adversarial regularized learning. In Proceedings of the Twenty-Ninth Annual ACM-SIAM Symposium on Discrete Algorithms, pages 2703-2717. SIAM.

---

> > ### Comment · Reviewer_wNLY · 2021-09-02
> > **Response to rebuttal**
> >
> > First, I'd like to thank the authors for their thorough response to my review. The additional context on the state-of-the-art on Markov games is welcome since I am not an expert on Markov games (in particular, I had misunderstood the position of your work wrt [Zhao et al. 2021]).
> >
> > However, after reading the other reviews and discussing with the AC and the reviewers, I share some of their concerns.
> >
> > Though I agree that your main contribution --- the (dimension-free) linear convergence guarantee for optimistic-type MWU methods on entropy-regularized games --- is significant, I am not comfortable about the presentation and the positioning of this work. I concur with other reviewers in that implementing the required modifications would require another round of reviews, which is unfortunately not the case at NeurIPS.
> > Here are a few modifications that, to my mind, would be required.
> > - Clarify that the algorithms are not part of the contribution, and if possible, present your results on the standard versions of the algorithms or explain why introducing such variants is required (see discussion with Reviewer v4uF).
> > - Explicit and highlight the role of the strongly monotone regularization in getting a linear convergence result even without the uniqueness of the equilibrium, eg by at least making an analogy with the Euclidean case (given a bilinear min-max on the whole Euclidean space, adding a quadratic regularization ensures the existence of a unique equilibrium of the regularized problem and a linear convergence to this equilibrium).
> > - Overhaul the discussion on entropic regularization in games (since I am not an expert on this subject, I leave it to another reviewer or the AC to provide the appropriate references).

---

### Official Review · Reviewer_v4uF · 2021-07-15

**Rating:** 4
**Confidence:** 4

**Summary:**

In this work, the authors propose to apply a multiplicative extragradient method (under the label "preductive updates" (PU)) and a variant (optimistic multiplicative weight update) to the solution of entropy-regularized matrix games and two-agent tabular markov games.



**Limitations And Societal Impact:**

The authors should more clearly formulate the significance of their results.

**Main Review:**

The first part of the paper (pages 3-7) studies the application of a multiplicative form of the extragradient method and optimistic mirror descent to entropy-regularized matrix games.
Convergence results are derived, including a suggestion for the choice of the regularization weight in order to compute an approximate equilibrium of the unregularized problem.
Finally, a synthetic experiment with an i.i.d random payoff matrix is provided.

The second part of the paper (pages 7-9) studies the application of the same algorithm to the solution of entropy-regularized value iteration in tabular markov games.
Leveraging the results from the first part, the authors provide an end-to-end convergence result for this application, in the absence of no sampling noise.

While there has been a vast number of recent papers studying variants of the extragradient method to different multi-agent games, there could still be space for a paper studying
the explicit application to matrix games.
However, the work is presently unfocused in its contribution (new algorithm vs analyzing an existing algorithm, theoretical novelty vs practical improvement) that I have to recommend rejection for now.

Suggestions for improvement:
Since algorithm 2 is the specialization of a known algorithm and "PU" can be obtained from the extragradient method by replacing the additive gradient step with a multiplicative one, it seems that the motivation and introduction of these algorithms is excessive.
Since there are numerous existing theoretical results on extragradient-type algorithms, it would be helpful to more clearly point our differences to existing results.
Also, since the PU and OMWU seem to perform almost identically, it is unclear why the authors consider both variants.
Throughout the paper, there are no empirical comparisons to other methods. If the authors could provide more evidence regarding when and why to use the methods presented in the paper and how they compare to other approaches, this could greatly increase the potential impact of the paper.


Minor comments:
(1) Line 62: "which is extremely natural and popular for optimizing over probability simplexes" To me, such strong yet vague statements without supporting evidence do not seem suitable for a scientific publication.

(2) Proposition 2: This result has nothing to do with the choice of the inner optimization loop. Is this a new result?



**Time Spent Reviewing:**

2

---

> ### Author Response · Authors · 2021-08-10
> **Thanks for the valuable feedback and insightful questions. Point-to-point answers are provided below.**
>
> ### --- "The work is presently unfocused in its contribution (new algorithm vs analyzing an existing algorithm, theoretical novelty vs practical improvement).'', ""it seems that the motivation and introduction of these algorithms is excessive''
>
> We note that while extragradient update is a well-known algorithmic idea, applying it to entropy-regularized matrix games can lead to different update rules and hence different analysis, depending on the way that the composite structure of the objective function are handled. Therefore, we believe it is essential to explain the algorithm derivation in details, which makes the paper more accessible to non-experts. Our contribution therefore includes both developing new algorithms and establishing novel theoretical analysis --- which significantly improved upon previous analysis by at least polynomially on the size of the action spaces --- under milder conditions (see Appendix A for comparisons with related literature). Our contributions are summarized formally in Section 1.2, which include 1) almost dimension-free linear convergence to QRE of entropy-regularized matrix games; 2) last-iterate convergence to NE of unregularized matrix games without assuming the NE is unique as in prior literature; 3) almost dimension-free linear convergence to QRE of entropy-regularized Markov games.
>
>
> ### --- "Since the PU and OMWU seem to perform almost identically, it is unclear why the authors consider both variants''
>
> While PU represents the most natural form for extragradient updates, the main difference between these two is that the OMWU's update rule requires only one pair of gradient evaluation (namely computing $A\bar{\nu}^{(t)}$ and $A^T\bar{\mu}^{(t)}$) per iteration, while PU requires two pairs of gradient evaluation per iteration. Nonetheless, while PU doubles the number of calls to the sampling oracle, it also allows roughly as large as twice the learning rate compared with OMWU (cf. (6)). Besides, OMWU achieves no-regret learning and is hence more robust against adversarial opponents in the game setting (we will include regret analysis for OMWU in the final version), while PU is not no-regret. Therefore, PU and OMWU have different features that none is strictly dominated by the other, therefore, we choose to present both algorithms and discuss their respective convergence properties.
>
> ### --- Empirical comparison to other approaches
>
> To the best of our knowledge, this work is the first one that develops provably linear converging extragradient methods for entropy-regularized matrix games. Our paper is theoretical in nature, and we have allocated the comparisons to other approaches for solving matrix games in terms of theoretical guarantees in Appendix A of the original submission. We will also add more comparisons to related works in the Markov game setting, as discussed in the response to Reviewer wNLY. We will try our best to include more empirical comparisons in the final version.
>
> ### --- Line 62:
> Thanks for the comment. We will modify the statement in the revision.
>
> ### --- Proposition 2:
> If the soft Bellman operator can be evaluated exactly, then one can perform entropy-regularized value iteration as in detailed in (16). Proposition 2 demonstrates the $\gamma$-contraction property of value iteration which generalizes the original Bellman contraction property to the entropy-regularization setting.

---

> > ### Comment · Reviewer_v4uF · 2021-08-10
> > **Motivation for new extragradient variants**
> >
> > Thank you for the response to my critique. Regarding the first point, it is my understanding that the only difference of the first method (PU) proposed in the paper to that of [mertikopoulos et al.](https://arxiv.org/pdf/1807.02629.pdf) is that the extragradient-idea is only applied to the matrix-game-part, not to the regularizer. Is that correct?
> > If so, does this modification lead to different behavior (in theory or practice)? I did not see an explanation to this effect in the paper, which is why the introduction of yet two more variants of existing extragradient algorithms seemed unnecessary to me.

---

> > > ### Author Response · Authors · 2021-08-11
> > > **Response to Reviewer v4uf**
> > >
> > > We want to take the opportunity to emphasize that our work focuses on understanding the effectiveness of extragradient methods for solving the regularized problem (i.e. finding QRE) at a global linear rate, and how entropy regularization enables solving the unregularized problem (i.e. finding NE) at a relaxed condition than all prior works --- that is, without imposing the unique NE assumption.
> > >
> > > With the above in mind, the reviewer is right in that PU only applies extra-gradient idea to the matrix-game term, which is inspired by the fact that entropy regularization term operates on $\mu$ and $\nu$ separately. Singling out the regularization term fits the geometry of the regularized problem better, gives a more concise closed-form update rule and leads to a cleaner analysis as well.
> > >
> > > More technically, the regularized problem offers a much simpler analysis and a different optimization dynamic than its unregularized counterpart. For instance, entropy regularization implies unique QRE and allows us to establish global linear convergence (with almost dimension-independent rates) throughout the entire trajectory of the algorithm, as opposed to the two-stage argument (sublinear convergence followed by local linear convergence with dimension-dependent rates -- see Appendix A in the supplementary material) in previous literature of finding NE (e.g. the analysis of OMWU in Wei et al, 2021b). This allows us to bypass the unique NE assumption and derive last-iterate convergence for the unregularized problem.

---

> > > > ### Comment · Reviewer_v4uF · 2021-08-11
> > > > **Does the modification lead to different behavior (in theory or practice)?**
> > > >
> > > > Just to make sure I understand: Your answer to my question whether the deviation from the extra mirror descent presented in Mertikopoulos et al. leads to different behavior in theory or practice is the following?
> > > >
> > > > >Singling out the regularization term fits the geometry of the regularized problem better, gives a more concise closed-form update rule and leads to a cleaner analysis as well. <
> > > >
> > > > Does that mean that the same convergence result and the same practical convergence would be achieved by the method of Mertikopoulos et al. just with a slightly longer update rule or slightly messier convergence proof? Or is there a fundamental challenge in analyzing the original extramirror descent in the same way?
> > > >
> > > > I am sorry if this appears nitpicky, but the abstract reads: "We develop provably efficient extragradient methods to find the quantal response equilibrium (QRE)—which are solutions to zero-sum two-player matrix games with entropy regularization—at a linear rate"
> > > > If there are no significant differences to the original extra mirror beyond the aestetic considerations quoted above, then I don't think it's justified to make these minor modifications to an existing algorithm and frame the paper as "developing a new method".
> > > >
> > > > This is not at all to say that your theoretical results are irrelevant. On the contrary, I definitely believe that they have a place in a venue like NeurIPS. However, I find the way in which they are presently presented somewhat confusing, and in need of clarification before publication. I believe that clearly linking the theoretical results to an existing algorithm as opposed to introducing two minor variants without demonstrated improvement over the existing one will make the paper stronger and more impactful.

---

> > > > > ### Author Response · Authors · 2021-08-12
> > > > > **Response to Reviewer v4uf**
> > > > >
> > > > > Thank you very much for engaging in this very useful discussion thread with us! We are happy to hear that you find our theoretical results relevant and worth publication at a venue such as NeurIPS.
> > > > >
> > > > > The sentence you quoted indeed captured our answer, and you raised another interesting question on whether it is possible to achieve the same practical convergence by the method of Mertikopoulos et al. “with a slightly longer update rule or slightly messier convergence proof” — unfortunately we are not aware of any existing analysis that achieves this goal, and we suspect analyzing the method of Mertikopoulos et al. might be somewhat similar to the situation when analyzing the vanilla OMWU for the unregularized problem, where we already see the analysis is significantly messier with somewhat weaker results (Wei et al 2021b). It might be possible that a drastically new proof strategy might allow a stronger theory for the standard extragradient mirror descent, but it is beyond the current paper. We believe our approach, as mentioned, suits the regularized problem better and goes beyond just aesthetic values.
> > > > >
> > > > > To address your concern about positioning our algorithm with related literature, we want to reassure you that we will definitely revise and clearly position our algorithm with existing ones, particular Mertikopoulos et al., and highlight the connections/improvements as this thread of discussions has illuminated.

---

### Official Review · Reviewer_1R9o · 2021-07-28

**Rating:** 7
**Confidence:** 3

**Summary:**

In this work, the authors aim to complete the recent line of research in last-iterate convergence guarantees for saddle-point optimization (applicable to game theory, GANs, etc.) with efficient methods that converge to an approximate Nash equilibrium *without* the equilibrium uniqueness guarantee. The key is to simply apply the previously analyzed methods (OMWU in particular) on a regularized version of the payoff. The regularization ensures uniqueness of what is called QRE, quantal response equilibrium, which can be used for finding an approximate Nash equilibrium. As a bonus, the authors show that their results apply to Markov games too.

**Main Review:**

I believe this work has enough significance to be published at this conference. The main contribution is getting rid of the uniqueness assumption in finding approximate Nash equilibria in bilinear games, since last-iterate convergence results for the constrained setting (with that assumption) have been derived in recent prior work. There is no algorithmic novelty, the methods are the same as in previous work. The key difference is using entropy regularization to ensure uniqueness, then convert to Nash, the desired outcome. The work might seem incremental in terms of techniques, methods, etc. but it still improves upon prior work by getting rid of a nontrivial assumption and that alone is sufficient for me to accept the paper.

**Time Spent Reviewing:**

8

---

> ### Author Response · Authors · 2021-08-10
> **Thank you for your positive assessment!**
>
> We want to mention that to the best of our knowledge, the proposed extra-gradient algorithms for solving entropy-regularized matrix and Markov games have not been developed nor analyzed before in the literature, and are indeed novel. In addition, the convergence rates obtained in this paper improved upon previous analysis by at least polynomially on the size of the action spaces (please see Appendix A in the submission), which we believe is essential in practice.

---

### Decision · Program_Chairs · 2021-09-28

**Decision:**

Accept (Poster)

**Comment:**

This paper treats the convergence of a regularized variant of the multiplicative weights and optimistic multiplicative weights update (MWU and OMWU respectively) in zero-sum games. A second part of the paper concerns the applications of these methods to zero-sum Markov games.

This paper was extensively discussed by the committee, and the reviewers identified both strong and weak points in the paper. On the positive side, the authors' linear convergence result for the regularized MWU/OMWU methods seems to be new (at the very least, the committee members were not aware of an equivalent result in the literature). On the negative side, the presentation and positioning of the paper left a lot to be desired, especially with regard to the similarity of the proposed regularized methods to other existing methods – and, in particular, running MWU/OMWU on a regularized game.

The main point of contention is as follows: if one considers an entropic regularization of the underlying game, it is immediate to see that the game's set of Nash equilibria is replaced by a unique quantal response equilibrium - or, rather, a logit equilibrium (as QRE are called in the context of entropic regularization; the term "Nash distribution" is also sometimes used). Thus, by focusing on QRE, the authors are essentially side-stepping the unique equilibrium requirement: they do not require equilibrium uniqueness, but they prove convergence to a perturbed equilibrium, not a Nash equilibrium. This point is crucial for the proper positioning of the paper, but it is not made clear by the authors.

Building on this, given that logit equilibria are _de facto_ interior, and given that the regularized game is strongly monotone, it is natural to expect a geometric rate of convergence for mirror descent / mirror-prox methods. [This, after all, is an entropic variant of the standard Tikhonov regularization approach.] Of course, the fact that this may be a "natural" result, does not subtract from its merit: however, a much clearer presentation of the topic would be expected in order to position these results in the proper context. [Further compounding the issue is that the authors' algorithm is not _exactly_ MWU/OMWU ran on the regularized game but closely related to it - and, if anything, this raises the question of why the authors' chose one variant over the other (especially since the former approach would seem simpler to analyze)]

Finally, regarding the applications to Markov games and $Q$-learning: one important limitation is that the paper operates in the "full-information" framework (with respect to individual player knowledge, not global one). While this assumption is meaningful from an "offline" viewpoint, it is harder to justify in the online setting where such algorithms are typically deployed (since only obtained payoffs are observed and counterfactual reasoning is not feasible in general). The authors touch on this issue briefly in the conclusions section, where they comment on the relevance of two-time-scale algorithms for the $Q$-learning problem: in this regard, the authors might want to check the 2005 paper of Leslie and Collins [1] and a follow-up by Coucheney et al. [2]. Both papers treat the problem of learning in normal-form games with entropic regularization and partial information, and the algorithms studied in both papers are very closely related to the regularized MWU algorithm studied by the authors; more to the point, [1] considers a two-time-scale variant that seems to do what the authors suggest in the conclusions section (modulo the extra-gradient part).

To sum up, the extent of the revision required to bring the paper's contributions into focus led the committee to the conclusion that the paper should go through another round of review before being considered again for publication. The decision to reject the current version was taken in this light, and I would strongly encourage the authors to resubmit a suitably revised version of the paper at the next opportunity once they have addressed the committee's concerns.

[1] D. S. Leslie and E. J. Collins, Individual Q-learning in normal form games, SIAM Journal on Control and Optimization 44 (2005), no. 2, 495–514.

[2] P. Coucheney, B. Gaujal, and P. Mertikopoulos, Penalty-regulated dynamics and robust learning procedures in games, Mathematics of Operations Research 40 (2015), no. 3, 611– 633.

**Consistency Experiment:**

NeurIPS has a long history of experimentation. In 2014, NeurIPS ran an experiment in which 10% of submissions were reviewed by two independent committees to quantify the randomness in the review process. This year, we repeated a variant of this experiment to see how the quality of the review process has changed over time.  This paper was part of the experiment and was therefore assigned to two committees (consisting of reviewers, an Area Chair, and a Senior Area Chair) that reached independent decisions.  If both committees made the same recommendation, this recommendation was followed. If a single committee recommended acceptance, the paper was accepted (with the exception of a few cases in which the other committee identified what we considered a fatal flaw, e.g., an error in a key result).

This copy’s committee reached the following decision: **Reject**

The other committee assigned to the paper recommended **Accept (Poster)**.  You can find the other set of reviews, along with any follow up discussion with the authors here:
https://openreview.net/forum?id=ZIyj0E58vzlo